# The Role of Selenium on the Formation of Spheroidal Graphite in Cast Iron

**Gorka Alonso [1,\*], Doru Michael Stefanescu [2,3], Edurne Aguado [1] and Ramon Suarez [1,4]**

[1]  AZTERLAN, Basque Research and Technology Alliance (BRTA), 48200 Durango, Spain; eaguado@azterlan.es (E.A.); rsuarez@azterlan.es (R.S.)

[2]  Department of Materials Science Engineering, The OHIO State University, Columbus, OH 43210, USA; stefanescu.1@osu.edu

[3]  Department of Engineering, The University of Alabama, Tuscaloosa, AL 35487, USA

[4]  Veigalan Estudio 2010 S.L.U., 48200 Durango, Spain

\*  Correspondence: galonso@azterlan.es; Tel.: +34-946-215-470

**Abstract:** Sulfur, an element that belongs to group 16 (chalcogens) of the periodic table, is an excellent promoter of nucleation substrates for graphite in cast iron. In ductile iron, sulfur favors a higher nodule count, which inhibits the risk of carbides and of microporosity. It is reasonable to expect that other elements from group 16, such as selenium or tellurium, play similar roles in the nucleation of graphite. The objective of this paper was to investigate the effect of selenium on the process of graphite formation. Thermal analysis cups were poured to evaluate the nodule count and size distribution. Some of the cups were not inoculated, while others were inoculated with a Ce-bearing inoculant, or with the Ce inoculant and additions of Se. Cross-shaped castings were also poured to quantify the microporosity regions by tomography. It appears that selenium additions modify the number and size of graphite particles, as well as the volume of microshrinkage. Direct correlations between these three parameters were found. Advanced Extensive Field Emission Gun Scanning Electron Microscope (FEG-SEM) techniques were used to identify the nature of the main nucleation compounds. Selenides, combined with Mg and rare earths, were observed to serve as nuclei for graphite. Their presence was justified by thermodynamics calculations.

**Keywords:** microporosity; sulfur; selenium; inoculant; nuclei





## 1. Introduction

It is well known that sulfur, an element that belongs to group 16 (chalcogens), plays a major role in the nucleation process of spheroidal graphite (SG) cast iron, usually in combination with other active minor elements, such as aluminum, calcium, cerium, or barium, introduced through the inoculation and/or spheroidization processes [1–4]. Many theories support the assumption of the heterogeneous nucleation on oxy-sulfides, Mg-Ca sulfides, or complex Mg-Ca-RE sulfides, as the main nuclei for spheroidal and compacted graphite [5–10]. These sulfides are among the most stable nonmetallic compounds. Similar observations were also made in gray cast irons [11,12].

A high sulfur content inhibits graphite spheroidization and increases dross formation. Alternatively, a low S level decreases the number of graphite particles and increases chilling. The harmful, or beneficial, effects of sulfur are related to the amount present before magnesium treatment, directly affecting graphite shape and nodularity. A minimum sulfur level of at least 0.005 to 0.008% is required after the spheroidization treatment to promote suitable nuclei for graphite precipitation and to reduce the risk of carbides. In SG irons, small amounts of iron sulfide can be added late in the process in order to achieve this critical content without any adverse effects on graphite nodularity.

Riposan et al. [13] showed that it was possible to use controlled sulfur additions during the postinoculation process to produce compacted graphite cast iron without

titanium additions. Sulfur was added as granular or powdery iron pyrites ($FeS_2$), producing inconsistent results because their fine mesh size A consistent control was achieved when briquetted iron pyrites were used. According to Suárez et al. [14,15], the late addition of small amounts of sulfur, in the form of pyrite granules ($FeS_2$) in high-silicon and high-carbon equivalent ductile irons, resulted in higher ferrite volume fractions, as well as larger nodule counts, promoting the formation of potent substrates for the nucleation of graphite. However, late sulfur additions can be detrimental to graphite morphology, setting a threshold level of about 0.006% S, with a fast degradation of nodularity for levels above 0.008% S. On the other hand, Nakae and Igarashi [16] proposed that the range from 0.010 to dwas the most desirable S content to produce SG iron castings in order to increase the SG nodule number. These researchers argued that spherical Mg-Ca sulfides were the main nuclei for SG, if the S content of the base melt was higher than 0.005%, while rectangular Mg-Si Al nitrides dominated when the S content was less than 0.0022% (Table 1).

**Table 1.** Influence of %S in the base iron on the nuclei of SG for Mg-treated iron, adapt from [16].

| %S in Base Iron | Nucleus Shape | dia. (μm) | Main Compounds | Other Compounds |
|---|---|---|---|---|
| 0.0022 | rectangular | 0.5–1.0 | (Mg,Si,Al) N | MgS, MgO, (Ca Mg) S |
| 0.0052 | spherical | 0.5–1.0 | (Mg,Ca) S | MgO, (Mg,Si,Al) N |
| 0.013 | spherical | 0.5–1.0 | (Mg,Ca) S | MgO, (Mg,Si,Al) ON |
| 0.050 | spherical | 1.0–2.0 | (Mg,Ca) S | MgO, (La,Ce,Nd) S |
| 0.072 | spherical | 1.5–5.0 | (Mg,Ca) S | MgO, (La,Ce,Nd) S |
| 0.083 | spherical/faceted | 1.5–5.0 | (Mg,Ca,Mn) S | MgO, (La,Ce,Nd) S |

It is also expected that other elements in the periodic table from the same group, such as selenium or tellurium, play direct roles in the nucleation of graphite. Although the three elements belong to the same group, 16, they present important differences in terms of density, crystal structure, melting point, thermal conductivity, and specific heat (Table 2). Thus, it is reasonable to assume that their behavior will be different in the process of the formation of graphite.

**Table 2.** Properties of some elements from group 16 of the periodic table.

| Properties | Sulfur | Selenium | Tellurium |
|---|---|---|---|
| Density ($kg/m^3$) | 1960 | 4790 | 6240 |
| Structure | orthorhombic | hexagonal | hexagonal |
| Melting Point (K) | 388.36 | 494 | 722.66 |
| Boling Point (K) | 717.87 | 957.8 | 1261 |
| Specific Heat (J/Kg·K) | 710 | 320 | 202 |
| Thermal Conductivity (W/k·m) | 0.269 | 2.04 | 2.35 |

The influence of tellurium on the formation of spheroidal graphite iron will be analyzed in future papers. This study will focus solely on the role of selenium. Most domestic selenium is produced as commercial-grade metal, averaging a minimum of 99.5% selenium, and is available in various forms. The global consumption of selenium during 2004 was stipulated at about 2700 metric tons, estimating the global end-use demand as follows: glass, 35%; chemicals and pigments, 24%; metallurgy, 23%; electronics, 10%; and other uses, 8% [17]. More than one-half of the metallurgical selenium was used as an additive (in amounts up to 1%) in cast iron, copper, lead, and steel alloys, improving the strength, ductility, casting, and forming properties, and even the resistance to corrosion in the case of magnesium-manganese alloys with additions of 0.3–0.5% Se [18].

In high-alloy steel castings, Se minimizes pinhole porosity. For stainless steels, the addition of selenium to the liquid produces selenide compounds, with some of the metallic elements appearing as inclusions in the steel matrix and improving the machinability. Kurka et al. [19] found Se in MnS-type inclusions that had a significant impact on their formability. A selenium content from 0.04 to 0.08% seems to affect the character, morphology, and dispersion of the nonmetallic inclusions, decreasing ductility and increasing

sensitivity to brittle fracture. Selenium compounds (selenides) are very unstable and, therefore, are mostly formed in solid steel on the surfaces of inclusions. When the steel solidifies, Se refines the grain structure, acting as a weak deoxidizer that contributes to better mechanical properties. In addition, the addition of small quantities of Se into the ladle may promote the formation of a finer and more equiaxed structure, with less directional differences in the properties [20].

There is not much evidence in the literature about the influence of selenium in the formation of graphite in cast irons. It is assumed that selenium is a surface-active element and, as such, it modifies the shape of graphite particles (from tiny flakes to nodules), which directly affects the mechanical properties. This theory was disputed by Horie [21], who analyzed the negative effects of tellurium and selenium on the formation of SG, and demonstrated that, by increasing the addition of both elements, the residual magnesium content decreased, and the shape of graphite changed successively from spheroidal to vermicular, undercooled, and then flaky.

The purpose of this article is to investigate the role of selenium on the formation of spheroidal graphite, studying its influence on the cooling curves, nodule count, and size distribution, as well as on the apparition of microshrinkage. The work will also address the impact of Se on the nucleation process through the precipitation of a new type of nonmetallic inclusions (selenides).

## 2. Materials and Methods

The iron was produced in a foundry in a 12-ton 7000 Kw induction furnace. The charge materials included: 5400 kg of steel scrap (0.01% C, 0.02% Si, 0.4% Mn, 0.02% P, 0.01% S, and 0.02% Cu), and 6600 kg of returns (3.76% C, 2.45% Si, 0.22% Mn, 0.04% P, 0.005% S, 0.07% Cu, and 0.022% Ti). The silicon level was adjusted by the addition of FeSi75 ferrosilicon (75.09% Si, 1.49% Al, and 0.77% Ca). The carbon level was corrected with synthetic graphite (54 kg). The melt was treated by the sandwich-method with Fe-Si-Mg alloy (45% Si, 5.5% Mg, 2% Ca, 2.28% RE) to spheroidize the graphite. The chemical compositions of the experimental heats are presented in Table 3. In addition to the elements listed in the table, the melt contained 0.06% Cr, 0.004% Sn, 0.006% Al, 0.010% Ce, and 0.004% La.

**Table 3.** Chemical compositions (% mass) of experimental cast irons.

| Heat | C | Si | P | S | Mg | Mn | Cu | Ti |
|------|------|------|-------|-------|-------|------|------|-------|
| 1 | 3.55 | 2.39 | 0.014 | 0.003 | 0.034 | 0.55 | 0.14 | 0.020 |
| 2 | 3.54 | 2.42 | 0.016 | 0.005 | 0.032 | 0.4 | 0.19 | 0.022 |
| 3 | 3.58 | 2.40 | 0.015 | 0.005 | 0.032 | 0.4 | 0.21 | 0.021 |

A series of standard thermal analysis (TA) cups, and cross-shaped castings (Figure 1), were poured from the melts, some not inoculated, while others were inoculated with a Ce-bearing inoculant (1.83% Ce, 0.95% Al, 0.91% Ca), or with the Ce inoculant with the addition of Se. Both selenium and the inoculant were added in a 1.3 kg hand-ladle before pouring the samples. Additions were 0.2% for Ce-inoculant (2.6 g/cup and 5 g for cross-shaped castings), and 0.0092% for selenium (0.12 g). The selenium was added as pure Se (99.9%). The elemental form is generally preferred for incorporation in ingots or castings because it melts faster [20]. A recovery of 66.6% was expected, the rest being lost by fume or slag. The TA cups were used to generate cooling curve information, as well as for microstructure analysis and for comparison with the nucleation in the cross-shaped castings used for the porosity measurements.

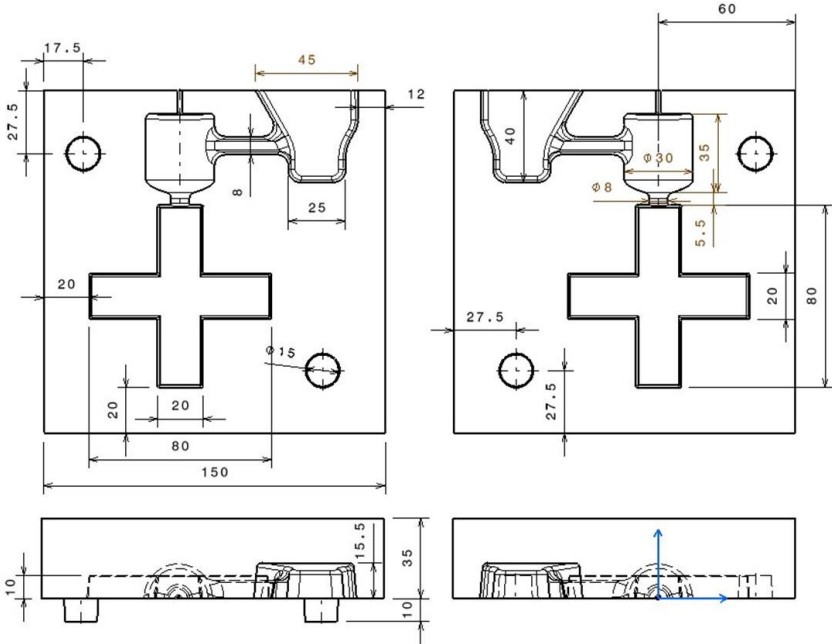

**Figure 1.** Cross shaped casting layout (dimensions in mm).

The cooling curves of the TA cups were recorded by Thermolan® system (V2, Veigalan Estudio 2010, S.L.U., Durango, Spain), and the cooling rates (first derivatives of the cooling curves) were calculated. An example of the output data is presented in Figure 2 for the noninoculated sample, and for the Ce-inoculant and the Se + Ce-inoculant. Information on undercooling, eutectic minimum temperature, recalescence, and the maximum cooling rate at the end of solidification ($CR_{max}$) were extracted from these curves. It can be seen that inoculation significantly increases the maximum cooling rate. Further discussion between the correlation between the porosity and $CR_{max}$ will be provided later in this paper.

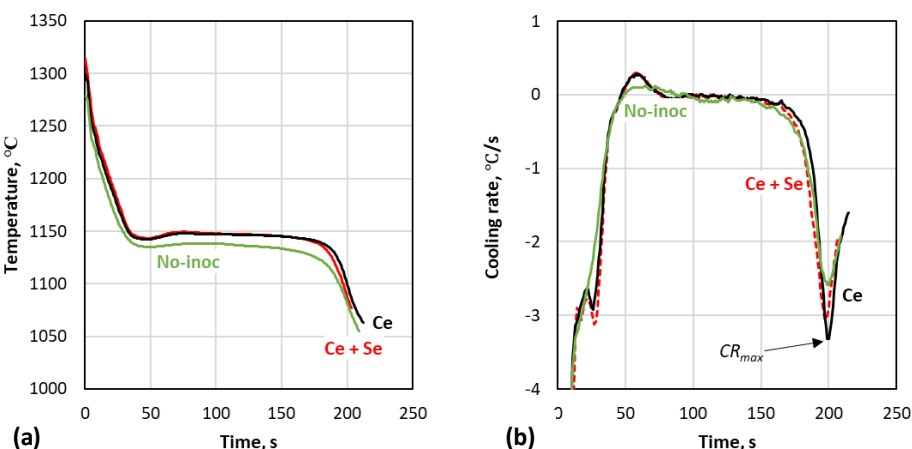

**Figure 2.** Cooling curves (**a**) and their first derivatives (**b**) from Heat 2 showing the effect of inoculation in increasing the maximum cooling rate, CRmax, at the end of solidification.

After cooling to room temperature, the cups were sectioned and prepared (ground and polished) for metallographic examination. A total of 10 different fields were taken for each sample in order to analyze the morphology of the spheroidal graphite by optical microscopy. Image processing was performed by means of the Image J software to determine the nodule count. The minimum size considered when counting graphite particles was a surface of 25 $\mu m^2$, which gives a diameter of 3.36 $\mu m$.

In order to identify the possible nucleation sites, an Ultra PLUS Carl Zeiss SMT (ZEISS, Thornwood, NY, USA) (0.8 mm resolution at 30 kV) in the STEM mode was used, in combination with an X-Max 20 Oxford Instruments EDX detector ( Oxford Instruments, Abrington, UK) with a resolution of 127 eV/ mm$^2$. The most advanced FEG-SEM techniques, such as spectrums, mappings, and line scans, were applied to analyze the main elements present in the inclusions, and to estimate the type of compounds that can act as nuclei for graphite.

The formation of Se-compounds was verified by the commercial software, FactSage6.41, whose theoretical basis is the equilibrium and phase transformations for the minimization of the Gibbs free energy. Two different calculation modules, "Equilib" and "Reaction", were used to determine the precipitation of possible compounds. The "Equilib" module determines the chemical composition of compounds when the elements react partially or totally to reach a state of chemical equilibrium under the chosen composition and temperature conditions (Figure 3). The "Reaction" module determines the change in the extensive thermodynamic properties, such as enthalpy, entropy, or specific heat to simple species or chemical reactions. It is seen that MgSe and La$_2$Se$_2$ are stable solids at temperatures of 1500 °C, and that Ce$_2$C$_3$ forms at 1200 °C. It is reasonable to assume that these compounds can act as nuclei.

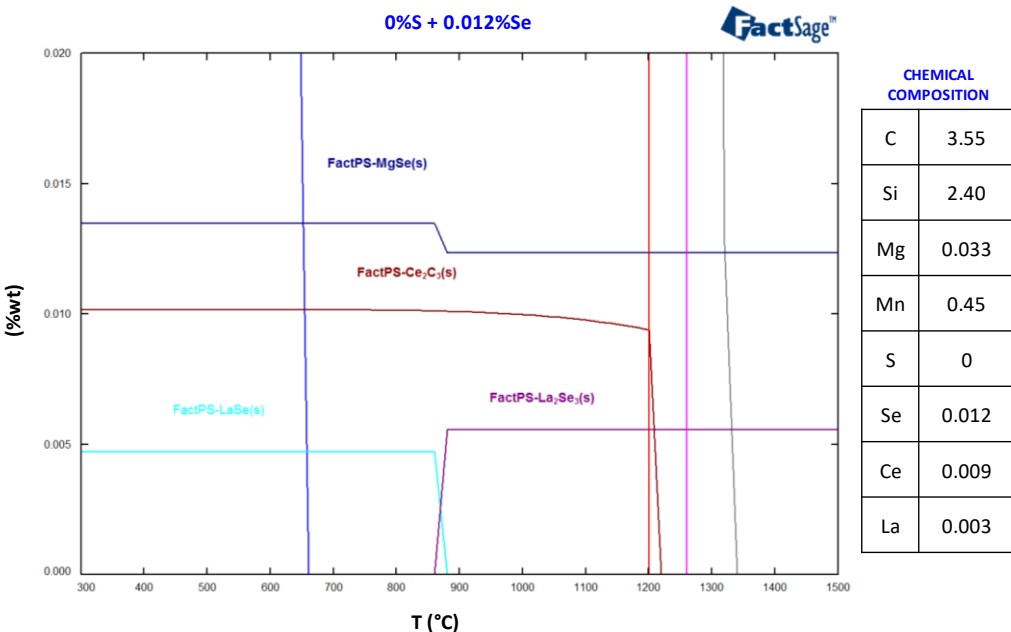

**Figure 3.** Stability of selenides according to FactSage, between 300 and 1500 °C, assuming the initial chemical composition listed in the figure.

X-ray computer tomography was used to evaluate porosity. This method has been proven to be accurate on SG iron samples by Borgs and Stets [22], who compared the tomography and the metallographic sections. In this work, tomographic inspection was performed on a YXLON equipment Mod. Y.CT Compact 450 kV and 1.5 mA (YXLON International X-Ray GmbH, Hamburg, Germany). The cross-shaped castings were sectioned perpendicular to the vertical axis, and the sections were analyzed. The distance between the sectioning planes was 1 mm, with a pixel size of 0.17 mm. A total of 83 sections per sample were produced. As an example, 4 of the 83 cuts on the sample with 1.8 Ce inoculant are shown in Figure 4.

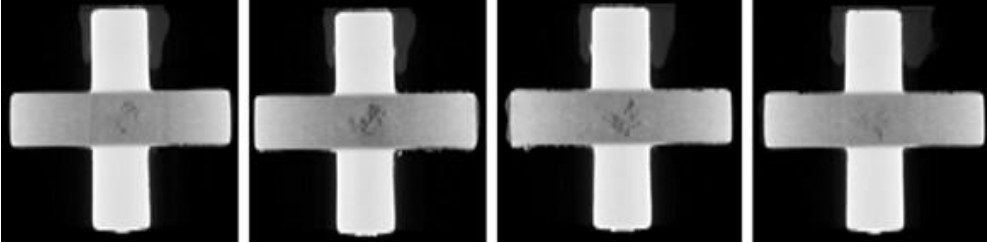

**Figure 4.** Example of tomography sections for a cross-shaped casting inoculated with Ce-inoculant; volume of internal defect: 101.82 mm³.

After the radiation of the sample, one file per plane was generated and then rebuilt into a 3D model using VG Studio Max 2.0. By means of this software, the total volume of the sample was calculated. By applying techniques based on contrast and grey levels analysis, the volume of defect is identified and evaluated.

## 3. Results and Discussion

### 3.1. Correlation between Cooling Curve Parameters, Nodule Count and Porosity

The experimental results are summarized in Table 4. It can be seen that, as expected, the nodule count increased considerably because of Ce inoculation. The Ce-Se combination was the most efficient, producing the highest number of nodules in all three heats.

**Table 4.** Experimental results showing the effect of inoculant and Se additions on formation of microshrinkage and cooling curve parameters.

| Heat | Inoc. | Nod/mm² | $T_L$ °C | $TE_{min}$ °C | $\Delta T_{recal}$ °C | TS °C | $CR_{max}$ °C/s | Microshrinkage mm³ |
|------|-------|---------|----------|---------------|-----------------------|-------|-----------------|---------------------|
| 1 | 1.8 Ce | 255 | 1146.8 | 1141.3 | 5.6 | 1118.6 | 3.80 | 11.22 |
|   | 1.8 Ce + Se | 377 | 1149 | 1147.4 | 2.9 | 1125 | 3.88 | 22.26 |
| 2 | none | 140 | 1135.9 | 1134.1 | 3.5 | 1104.6 | 2.44 | 148.46 |
|   | 1.8 Ce | 270 | 1148.2 | 1142.5 | 5.7 | 1125.1 | - | 2.85 |
|   | 1.8 Ce + Se | 333 | 1152.2 | 1144.5 | 7.8 | 1127.3 | 3.38 | 0 |
| 3 | none | 216 | 1139.7 | 1135.5 | 4.1 | 1106.1 | 2.58 | 272.82 |
|   | 1.8 Ce | 368 | 1148.5 | 1141.3 | 7.2 | 1125.4 | 3.32 | 101.82 |
|   | 1.8 Ce + Se | 386 | 1149.6 | 1143.1 | 6.5 | 1124.5 | 3.06 | 6.50 |

This improvement in nodule count translates, in most cases, to an important reduction in the size of the microshrinkage. As shown in Figures 5 and 6a, as the nodule count increases, the amount of porosity decreases. The addition of Se appears to emphasize this tendency, except for Heat 1, where the samples inoculated with 1.8 Ce + Se present a microporosity of about a factor of two more than those inoculated only with Ce. This difference can be attributed to the bad behavior of the feeder, or to a poor performance of Se. Because of the low value of sulfur in this sample (0.003% S vs. 0.005% S in the other ones), which has already been shown as an excellent promoter of potent substrates for the nucleation of graphite [5,8,10], graphite expansion may become insufficient to compensate the solidification shrinkage, increasing the risk of microporosity formation.

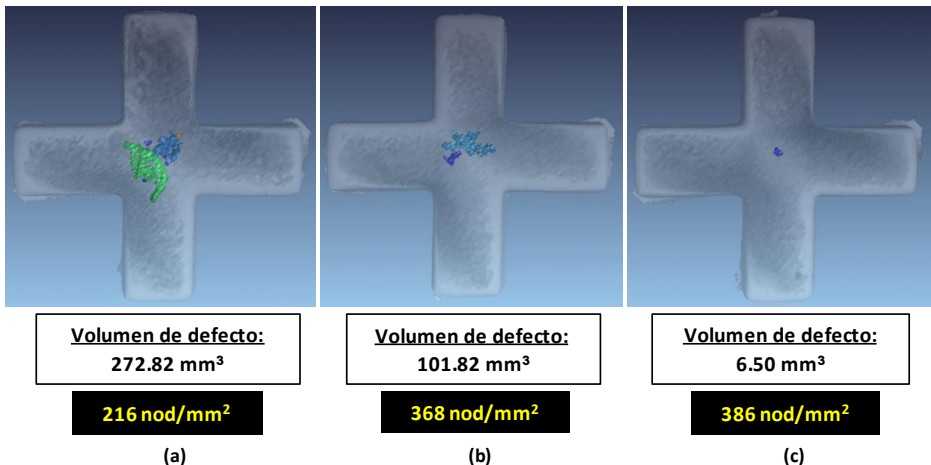

**Figure 5.** Evaluation of microporosity by tomography in cross-shaped castings from Heat 3: (**a**) non-inoculated; (**b**) inoculated with Ce-inoculant; (**c**) inoculated with Ce-inoculant + Se.

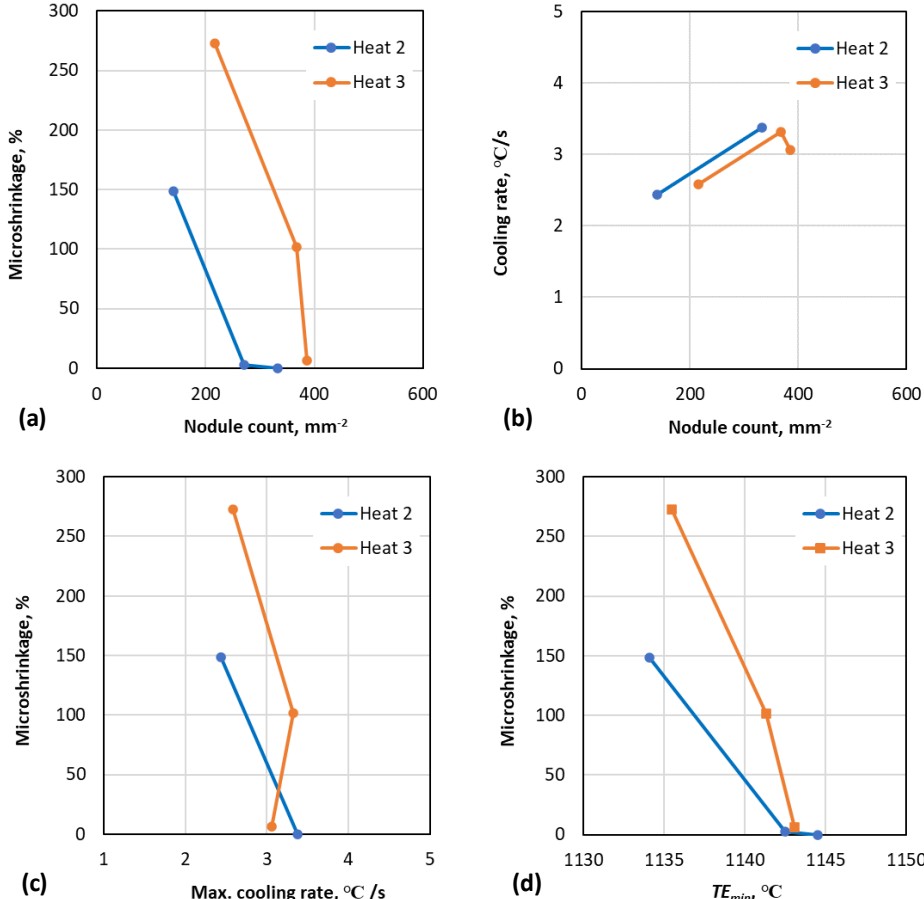

**Figure 6.** Correlation between various measured parameters and microshrinkage; the variable on the abscise reflects the sequential left-to-right change not inoculated-inoculated-inoculated + Se: (**a**) correlation between microshrinkage and nodule count; (**b**) correlation between nodule count and cooling rate; (**c**) correlation between microshrinkage and the maximum cooling rate; (**d**) correlation between microshrinkage and minimum eutectic temperature.

A good correlation is also seen between the nodule count and $CR_{max}$. However, the use of $CR_{max}$ in differentiating between Ce-inoculated and (Ce + Se)-inoculated is not conclusive as we only have one data point. Similarly, while $CR_{max}$ exhibits a clear increase

with inoculation (Figure 6b), it appears to be a less precise predictor of the differences in microshrinkage between the inoculated irons (Figure 6c). This is due, at least in part, to the inaccuracies resulting from Se assimilation in the melt during inoculation in the hand ladle. To reach a definitive conclusion on this issue, more research data are required.

Finally, as shown in Figure 6d, a higher $TE_{min}$ indicates a lower microshrinkage.

### 3.2. Size Distribution of Graphite

Ten images per sample were analyzed to characterize the size distribution of spheroidal graphite. An example of measured graphite nodule histograms for cups with and without inoculation (Ce-inoculant) is shown in Figure 7. The experiments indicate that the nucleation of SG can follow a monotonic trend (only one maximum on the size distribution curve) or can exhibit several nucleation waves (several maxima). Chisamera et al. [23] have shown that complex FeSi inoculants with Ca, Ce, S, and O extend graphite nucleation through the end of eutectic solidification. A bimodal volume size distribution of graphite nodules (a set of small nodules coexisting with a near normally distributed set of large nodules) was observed by Lekakh et al. [24] and suggest a link between the second nucleation wave and lower microporosity. While none of the TA cups in this research produced a second nucleation wave (two maxima) in the size distribution curves, a clear difference was observed in the microstructures and distribution curves for the inoculated irons as compared with the noninoculated. The lack of inoculation produces a more uniform distribution spread over different sizes of graphite, in addition to the bad shape parameters of spheroidal graphite (low roundness and high aspect ratio), and lower nodule count.

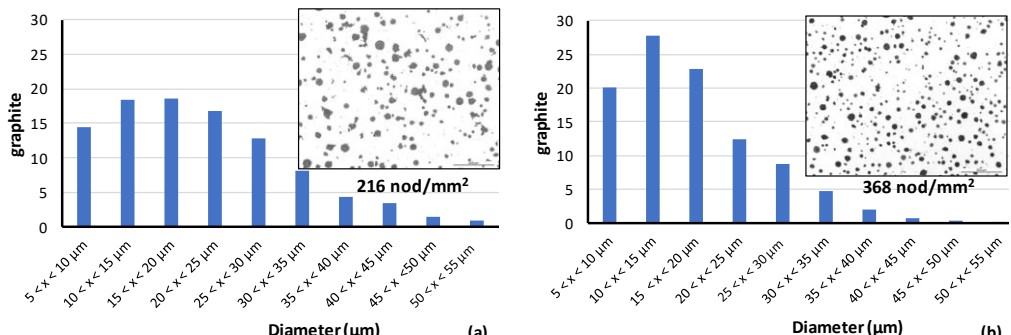

**Figure 7.** Unetched micrographs and graphite size distribution for samples from Heat 3 non inoculated (**a**) and inoculated with Ce-inoculant (**b**).

The addition of Se considerably modifies the size distribution of graphite, moving the formation of graphite to the left, generating finer graphite. Indeed, as shown in Figure 8, the addition of Ce-inoculant (1.8 Ce) produces a distribution with 55% of the nodules in the range of a 5–20 µm diameter, vs. 66% for the sample where (Ce + Se)-inoculant was added. This finer graphite generation, assumed to form at the end of solidification [25], could better counteract the austenite contraction in the last stages of solidification, and may be responsible for the decrease in the microshrinkage formation (Figure 5).

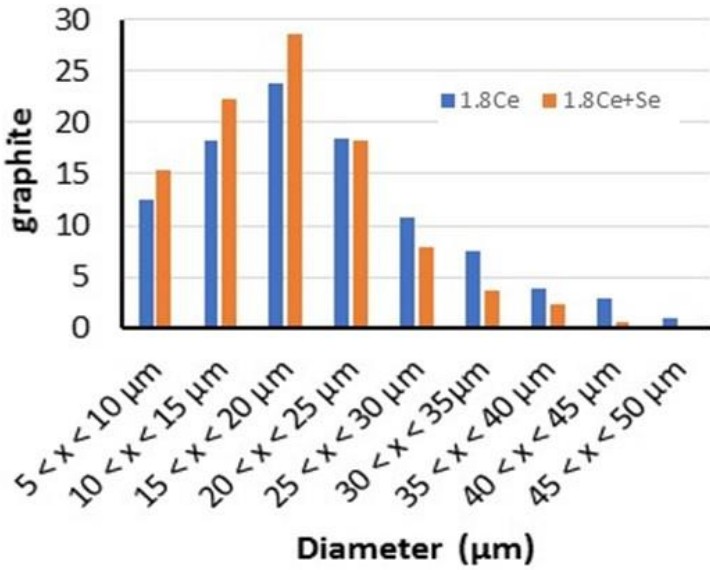

**Figure 8.** Influence of the addition of Se on graphite size distributions for TA cups from Heat 2.

### 3.3. Thermodynamics Calculations

According to the previous results, it appears that selenium contributes favorably to the formation of graphite. It is expected that this element forms different nonmetallic inclusions that can act as nuclei for graphite.

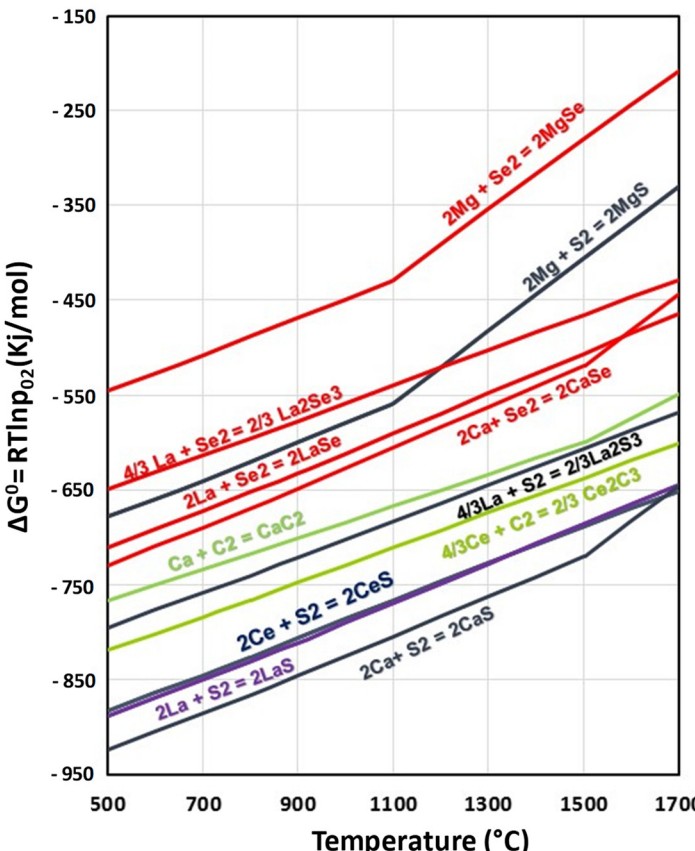

**Figure 9.** Standard free energy of formation of some selected sulfides and selenides.

The possible precipitation and stability of these compounds was analyzed through an Ellingham's diagram (Figure 9) based on the calculations from the FactSage database

(V6.4, GTT Technologies, 52134, Herzogenrath, Germany). This diagram, which shows the dependence of the stability for compounds as a function of temperature [26], reveals the formation of several thermodynamically stable Se compounds (selenides), such as CaSe, LaSe, and MgSe (it is somehow surprising that no Ce selenide was detected). Ca-selenides seem to be the most stable of all the selenides. However, sulfides exhibit, in most cases, lower standard free energy formation ($\Delta G^0$). Thus, it is expected that Mg, Ca, or RE prefer to combine with sulfur rather than with selenium. Depending on the content of Se and S, selenides can coexist with their corresponding sulfides. In the case of Ca, for example, the melting point for CaS is 2526 °C vs. 1408 °C for CaSe, so both can appear together if the temperatures are not very high.

### 3.4. Nature of Nuclei

An exhaustive SEM analysis of the type of nonmetallic inclusions that can act as nucleation sites for graphite shows that the main nuclei in the inoculated irons were made by rounded sulfides and polygonal Mg-Si-Al nitrides, as is summarized in Table 5. Both types of inclusions can appear alone (Figure 10a), or in the same graphite aggregate (Figure 10b).

**Table 5.** Main inclusions detected in the graphite acting as nuclei.

| Heat | Inoc. | No. Nuclei | Oxides | Sulfides | (MgSiAl) N | Ti (CN) | Graph. with RE | Graph. with Se |
|------|-------|-----------|--------|----------|-----------|---------|---------------|---------------|
| 1 | 1.8 Ce | 20 | 4% | 40% | 33% | 25% | 25% | 0% |
|   | 1.8 Ce + Se | 20 | 0% | 57% | 30% | 13% | 50% | 60% |
| 2 | 1.8 Ce | 20 | 7% | 46% | 36% | 11% | 35% | 0% |
|   | 1.8 Ce + Se | 20 | 3% | 47% | 29% | 21% | 50% | 50% |

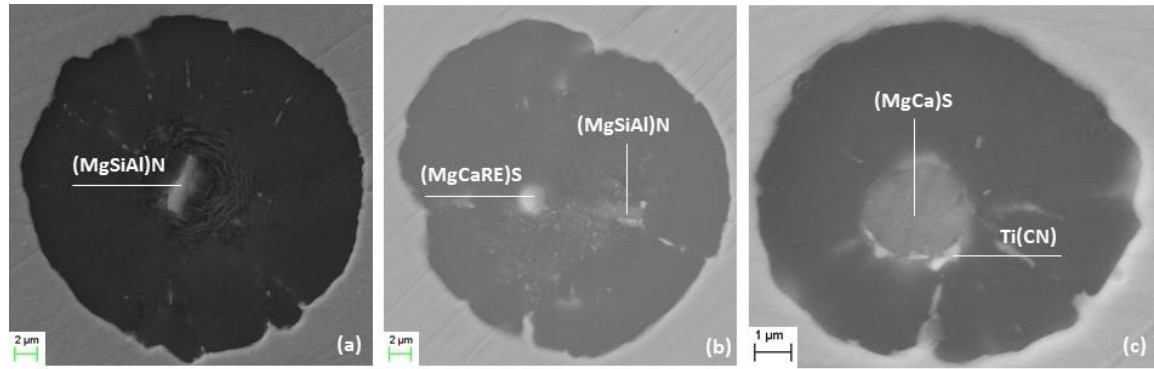

**Figure 10.** Different nonmetallic inclusions acting as nucleation sites for the graphite: (**a**) a complex Mg-Si-Al nitride; (**b**) a combination of (MgSiAl) N + (MgCaRE) S; (**c**) a small Ti (CN) growing on a big Mg-Ca sulfide.

Some oxides were also found. The formation of cubic Ti carbides is determined by the content of Ti in the base melt (>0.020% Ti) [3,27,28]. They are never found isolated and need another inclusion (complex nitride or sulfide) to nucleate (Figure 10c).

The presence of RE is relevant, particularly in samples with the addition of Se, where they are always detected as sulfides (Figure 10b), or selenides (Figure 11), which verifies the great affinity between S and Se and RE (mainly La and Ce). Their presence seems to be linked to the Mg treatment(45%Si, 5.5%Mg, 2%Ca, 2%RE) used for the spheroidization process. The detection of Se acting as a nucleation site for graphite in samples inoculated with 1.8 Ce + Se, which are characterized by a higher nodule count, is evident (>50% cases); thus, the important role that this element plays in the formation of graphite is clear. The theoretical formation of these Se compounds was justified previously by thermodynamic calculations.

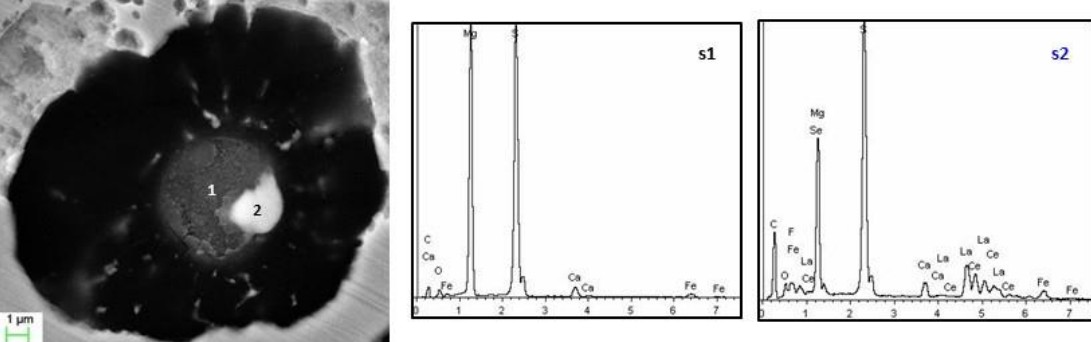

**Figure 11.** SEM images of a graphite nucleating on a selenide, and corresponding WDX/SEM spectrums, at different positions: position 1 − (Mg,Ca) S; position 2 − (Mg,Se,Ca,Ce,La) S.

These selenides can act as direct nucleation sites for graphite (Figure 11), or as nuclei for other nonmetallic inclusions (usually a complex of Mg-Si-Al-N). An example is illustrated in Figure 12, where an analysis, using X-ray concentration graphs, reveals that Mg, Si, Al, and N show composition peaks at the same position. Then, S, Se, and Ca present a clear coincidental maximum. Thus, the nucleus of this graphite seems to be formed by a big polygonal Mg-Si-Al nitride (≥10 μm) that has nucleated on two small Ca sulfides and selenides, demonstrating the coexistence of both compounds, as discussed earlier.

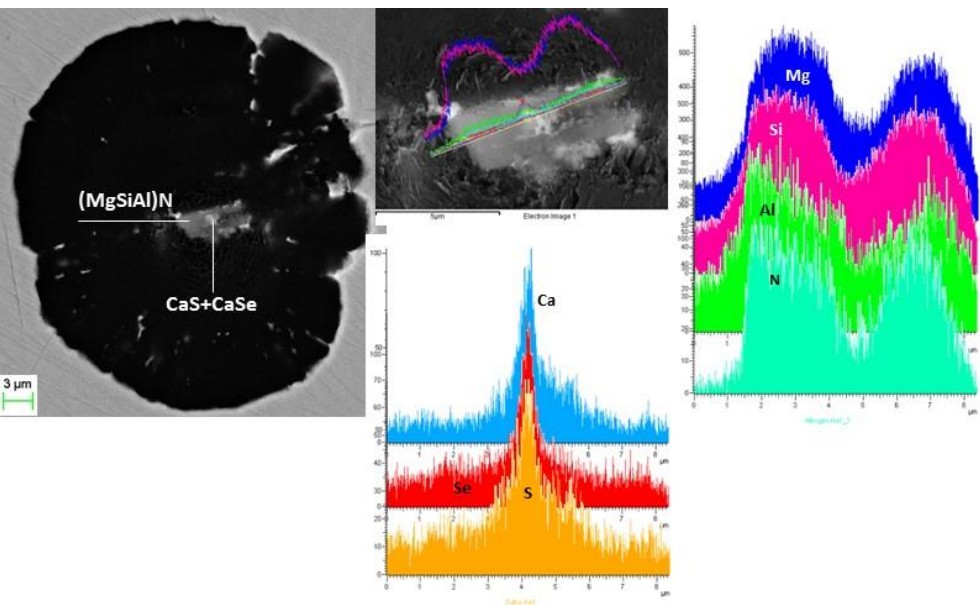

**Figure 12.** Determination of nuclei in function of X-ray concentration line scan graphs.

## 4. Conclusions

Correlations between the microshrinkage (shrinkage porosity) and some metallographic and thermal analysis parameters of spheroidal graphite irons with 4.3% carbon were evaluated on thermal analysis cups and cross-shaped castings poured from an industrial batch of Mg-treated iron. A commercial inoculant, rich in Ce and some additions of pure Se, was added in the hand-ladle before pouring the samples. The influence of these additions on the process of the formation of graphite was also studied.

As expected, the inoculation process considerably improved all the parameters of the cooling curves, as well as the nodule count, drastically decreasing the apparition of microporosity. It was found that the addition of selenium increases the eutectic minimum temperature and the solidus temperature, as well as the number of nodule spheroids. Selenium modifies the size distribution of graphite, producing finer graphite that seems

to nucleate at the end of solidification, reducing substantially, in most cases, the volume of microshrinkage. This behavior is explained by the formation of some Se compounds (selenides) that can act as excellent nuclei for graphite. Thermodynamics calculations and a complete SEM analysis supported this assertion.

**Author Contributions:** Conceptualization, G.A. and D.M.S., methodology, G.A.; software, G.A. and E.A.; validation, D.M.S., G.A. and R.S.; formal analysis, G.A.; investigation, G.A.; resources, G.A. and D.M.S.; data curation, G.A. and D.M.S.; writing—original draft preparation, G.A.; writing—review and editing, D.M.S. visualization, G.A.; supervision, R.S.; project administration, R.S. All authors have read and agreed to the published version of the manuscript.

**Funding:** This research received no external funding.

**Institutional Review Board Statement:** Not applicable.

**Informed Consent Statement:** Not applicable.

**Data Availability Statement:** Not applicable.

**Acknowledgments:** The authors would like to acknowledge Diputacion Foral de Bizkaia for supporting this research.

**Conflicts of Interest:** The authors declare no conflict of interest.

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
