# Peer review of "The Role of Selenium on the Formation of Spheroidal Graphite in Cast Iron"

_metals, doi:10.3390/met11101600_

Round 1
Reviewer 1 Report
This manuscript deals with the role of selenium on the formation of spheroidal graphite by studying the influence of selenium on the cooling curves, nodule count and size distribution, the apparition of micro shrinkage and the nucleation process.
The ideals are interesting and the findings are worthy of publication.
Also, there have some points must be clarified.
- In the context, many “Error! Reference source not found” appeared which shown that the wrong reference number! Please check them carefully!
- In Table 4, the effect of Se additions on formation of microshrinkage were given. Although the tendency of porosity decreasing are same, the porosities are very different in the volume between heat 2 and heat 3 with the same Ce addition of 1.8%. What is the reason?
- It must be careful that the correlation between the microstructural parameters and CRmax given in Figure 6 for very limited TA data!
Author Response
- In the context, many “Error! Reference source not found” appeared which shown that the wrong reference number! Please check them carefully!. I am afraid that this is a problem with the version of word or something similar, of your computer. I have downloaded the manuscript reviewed directly from the website of Metals, and I have not found any message of error or any problem with the references. In any case, and in order to facilitate the reading, I have also attached a pdf version of the final paper.
- In Table 4, the effect of Se additions on formation of microshrinkage were given. Although the tendency of porosity decreasing are same, the porosities are very different in the volume between heat 2 and heat 3 with the same Ce addition of 1.8%. What is the reason? The explanation would be related with differences of the filling of both samples. Although all the samples have been taken in the same conditions, differences in the filling times, or problems of turbulences could justify this contrast
- It must be careful that the correlation between the microstructural parameters and CRmax given in Figure 6 for very limited TA data! This is why the statement “To reach a definitive conclusion on this issue, more research data are required.” was already included in the text.

Reviewer 2 Report
Please see the attachment

Author Response
The paper offers a couple of interesting aspects for study the role of selenium on the formation of spheroidal graphite in cast iron. However, it is difficult to understand and follow the analysis because the legend (Error! Reference source not found) appears along the manuscript please refers to lines: 56, 62, 109, 128, 169, 171, 182, 189, 190, 205, 206, 212, 235, 249, 265, 266, 267, 271, 278, 279, 289, 291. Tables 1, 2, and 3 are presented without a previous explanation in the text, as well, as Figures 1, 3. For example in Line 110 “the elements listed in the table,” What Table? I am afraid that this is a problem with the version of word or something similar, of your computer. I have downloaded the manuscript reviewed directly from the website of Metals, and I have not found any message of error or any problem with the references. In any case, and in order to facilitate the reading, I have also attached a pdf version of the final paper.
It is an interesting item, however, I have the followings remarks:
Abstract The following text must be located in the introduction section: “Sulfur, an element that belongs to group 16 (chalcogens) of the periodic table, is an excellent promoter of nucleation substrates for graphite in cast iron. In ductile iron, sulfur favors higher nodule count, which inhibits the risk of carbides and of microporosity. It is reasonable to expect that other elements from group 16, such as selenium or tellurium, play a similar role in the nucleation of graphite.” That paragraph has been included in the abstract and most of the statements are repeated in slightly different format in the introduction. However, the following statement was added to the introduction: “an element that belongs to group 16 (chalcogens).”
Line 16, it's mentioned, “or with the Ce inoculant and additions of Se.” The results are presented only for one addition of selenium. This is not right. In Table 4 the results are shown for samples not inoculated, inoculated with Ce an inoculated with Ce+Se
Line 17, its mentioned, “It appears…” the results are not conclusive? The same statement is reported in Lines 191, 205, 245. The results are conclusive but in some cases, they are limited, and by this reason we would rather suggest than assert
- Materials and methods Line 104. What do you refer to as returns? Maybe iron scrap? Yes (feeders, channels, etc). This is a standard term in foundry technology that does not require additional explanations.
Table 3 must present the selenium in the chemical composition. It would be interesting, but we do not have that data. In any case, Table 3 shows the chemical composition of the different heats directly form the heating-pouring unit and the additions of Se were done in the hand-ladle.
It is not clear why three melts or castings with similar chemical compositions were used in the study. To generate more data and to obtain a repeatability in the results
How was the sulfur amount determined? By LECO
The inoculation method must be described. Inoculation method is already described in text “A series of standard thermal analysis (TA) cups and cross shaped castings (Figure 1) were poured from the melts, some not inoculated, while others were inoculated with a Ce-bearing inoculant (1.83% Ce, 0.95% Al, 0.91% Ca) or with the Ce inoculant and additions of Se. Both selenium and inoculant were added in a 1.3 kg hand-ladle before pouring the samples. Additions were 0.2% for Ce-inoculant (2.6 g/cup and 5 g for crossed shaped castings) and 0.0092% for selenium (0.12 g).”
It is confused the inoculation trials. An additional table must be included to clarify the trials developed for the inoculation variations. The inoculation variations are already included in Table 4
How was chosen the amount or amounts of selenium added? There is very little information in the literature about additions of Se in cast iron, so we have done several trials starting with small amounts in order to check their effect in the solidification process. This is standard research procedure.
Line 113, the text is part of table 3? No
Figures 2 and 3 must be in the results section. It could be, but, as they are used to explain the research methodology it is necessary to introduce them at an early stage.
I recommend a thermodynamic modeling section included in the materials and method section. It would be an option, but it has already been included in the results section. In other case, I would have to include also a section for tomography or SEM investigation.
Results and discussion
The work aims at the selenium effect on the nodule formation, only two small micrographs that not belongs to the selenium addition are shown in figure 7. The micrographs that allow presenting the results of the nodule count in table 4 must be presented. The information proposed by the micrographs is not significant because of samples with Ce or with Ce+Se present similar micrographs (the biggest difference is between samples inoculated or not). The most relevant information is on the nature of nucleation sites which is totally affected by Se additions and that section is developed in depth in the paper
The results in figure 2 must explain why heat 2 was chosen. Figure 4 presents results only for the inoculant addition, the inoculant with selenium must be included, for what heat? Figure 2 and Figure 4 are included in section of Materials and Methods only to understand better the methodology, not to explain the effect of selenium. Not all cases can be included. The influence of Se is explained in detail in section of results and discussions (Table 4, Figure 5, Figure 6)
Figure 6 presents results from heats 2 and 3 but is not clear what inoculant trial is presented. The purpose of the graphs is to demonstrate the role of inoculation. Each set of data includes the sequence not inoculated/ inoculated/ inoculated +Se. To clarify this issue the following statement was added in the caption:” the variable on the abscise reflects the sequential change not inoculated – inoculated -inoculated + Se”
Figure 3. The thermodynamic analysis is not related to the experimental trials. Why the chemical composition used excluded the S, Al, Ca, and oxygen elements? S is very important as was mentioned in the manuscript, Al and Ca are important elements of the inoculant and will form sulfides. Oxygen is presented in a low concentration in the melt and will form oxides inclusions also important to nucleation. Some elements such as Al, Ca or O have been excluded in order to simplify the calculations. The objective of that Figure 3 is only to demonstrate the possibility of formation of selenides from the point of view of thermodynamic
I considered that the thermodynamic analysis must be developed with a better approach to the experimental trials. That is not the objective of the paper. Thermodynamic analysis was only included to demonstrate the probability of formation of Se compounds from the point of view of thermodynamic, nothing more.
The thermodynamic properties (H, S, Cp) of the compounds involved in the analyses must be included. They are not relevant for the study
Use °C and wt % in Figure 3. Done
Lines 193 -194 “This difference can be attributed to a bad behavior of the feeder or to a poor performance of Se” The feeder was the same in all the trials as can be observed in Figure 1. Right, but it is well known that same feeders not always work in similar fashion.
Lines 194-195 “Because of low value of sulfur in this sample (0.003%S versus 0.005%S in the other ones),” However, the nodule count in heat 1 is higher than heat 2. Right. Probably in these cases the inoculation and Se additions have worked better.
Figure 9 shows the thermodynamic stability of the compounds formed. I consider that the results of Figure 3 with more elements included must provide more appropriate results than Figure 9. Fig 9 is an Ellingham’s diagram which can be done for different compounds, sulfides, oxides, nitrides, etc. In this case sulfides and selenides have been included because they are the most important nuclei for SG. Besides, many of elements of Fig. 3 can form compounds which are not stable being discarded
If A = A or Se, why appears C in Figure 9? To avoid confusion, the statement “, corresponding to the general reaction (2x/y)M + A2=(2/y)MxAy where A can be S or Se” has been removed from the caption.
Line 280, what is (5-2-2) ? It is the typical nomenclature for Mg treatment where 5 is the %Mg, 2 is the %Ca and the last 2, the % RE. However, to clarify 5-2-2 was replaced with Fe-45Si-5.5Mg-2Ca-2RE
It’s difficult to relate the SEM results with the text explanation. The quality of figures 10, 11, and 12 must be improved. From figure 12 it’s evident the relationship between the CaS (thermodynamically more stable) with the Se compound, this fact could be evidenced with a better thermodynamic analysis. We do not think that it is difficult to relate SEM results and text. The main nucleation sites are well described, summarized in a table (Table 5) and even related to inoculant and Se additions. Besides, their presence has been justified in three different ways, with spectrum, line scans and mapping. In the same way, the quality of the pictures is obtained directly from the SEM program and is good. The graphite is well defined, the formation of the nuclei is clear as well as the graphs described for line scan and mapping. Probably more thermodynamic calculations can be done to show the relationship between Ca and Se. We will take into account for the future

Reviewer 3 Report
Interesting research.
Redaction for the paragraph on lines 76-87 could be improved. For example, the idea on lines 82 and 83 is confusing.
Several messages, “Error! Reference source not found” appear on the text.
Some commentary about hot-spot formation in the centre of cross-shaped casting would be helpful for readers.
The definition of the thermal parameters used in the paper must be shown in Figure 2.
In FactSage, the "Reaction" mode must be where only pure substances are taken into account, and the “Equilib” module must take phases like slag, mattes, liquid or solid alloys. If that is true, please improve the redaction of lines 148-158. The start and the end of the solidification of alloy could also be indicated in Figure 3.
In Table 4, please mention the minimum size of nodules considered in the counting. I guess that it is 5 microns.
Would you please comment on why the porosity volume of cross-shaped casting is related to the volume of the TA cup?. For example, this porosity could also be related to the volume of the intersection section defined by design rules presented in the Casting Handbook. Microshrinkage values over 100% are in some way confusing.
Figures 7 and 8 do not need hundredths in % graphite.
In line 294, please comment over which size a nitride is considered to be big.
Author Response
Redaction for the paragraph on lines 76-87 could be improved. For example, the idea on lines 82 and 83 is confusing. Those sentences were included only to show a little bit the effect of additions of Se in the properties of other materials such as copper, lead or steel and we believe that it is well described
Several messages, “Error! Reference source not found” appear on the text. I am afraid that this is a problem with the version of word or something similar, of your computer. I have downloaded the manuscript reviewed directly from the website of Metals, and I have not found any message of error or any problem with the references. In any case, and in order to facilitate the reading, I have also attached a pdf version of the final paper.
Some commentary about hot-spot formation in the centre of cross-shaped casting would be helpful for readers. Porosity formation in hot spot is a well-known and documented phenomenon in the metalcasting literature. We believe it does not require further explanations.
The definition of the thermal parameters used in the paper must be shown in Figure 2. Done
In FactSage, the "Reaction" mode must be where only pure substances are taken into account, and the “Equilib” module must take phases like slag, mattes, liquid or solid alloys. If that is true, please improve the redaction of lines 148-158. The start and the end of the solidification of alloy could also be indicated in Figure 3. The information related to the modules of Factsage has been obtained directly from its operation manual. All the thermodynamic calculations could be studied in depth but it is not the objective of our study. In fact, there is a specific module for slags, but the information proposed here is only to facilitate reading of the paper and to support the results obtained by the SEM study. We can consider that suggestion for future publications.
In Table 4, please mention the minimum size of nodules considered in the counting. I guess that it is 5 microns. The statement “ The minimum size considered when counting graphite particles was a surface of 25 µm2, which gives a diameter of 3.36 µm “ was included in the text.
Would you please comment on why the porosity volume of cross-shaped casting is related to the volume of the TA cup?. For example, this porosity could also be related to the volume of the intersection section defined by design rules presented in the Casting Handbook. Microshrinkage values over 100% are in some way confusing. The porosity volume of cross-shaped casting is not related to the volume of the TA cup, it is related with the parameter of cooling curves (which give information about the metallurgical quality of the base iron) obtained from thermal cups. The microshrinkage values appears in Table 4 as volume (mm3) so the sentence “The final % of porosity is obtained by dividing the porosity volume by the total volume of the TA cup by 100” has been removed.
Figures 7 and 8 do not need hundredths in % graphite. Right. New graphs have been included
In line 294, please comment over which size a nitride is considered to be big. There is not a stablished size to determine if a nitride is big or small. The equivalence “≥10µm” has been introduced in the text

Reviewer 4 Report
The ABSTRACT section is well-structured. The INTRODUCTION section provide the necessary background information. The paper is structured properly (MATERIAL AND METHODS, DISCUSSION, CONCLUSIONS, REFERENCES, etc.). The METHODOLOGY is relatively well described. The body of paper describe the important RESULTS of the research. The CONCLUSION section succinctly summarize the major points of the paper. The list of REFERENCES is long and relatively well chosen.
Author Response
Thanks for your contribution
Round 2
Reviewer 1 Report
Some last comments were not answered!
- In Table 4, the effect of Se additions on formation of microshrinkage were given. Although the tendency of porosity decreasing are same, the porosities are very different in the volume between heat 2 and heat 3 with the same Ce addition of 1.8%. What is the reason?
- It must be careful that the correlation between the microstructural parameters and CRmax given in Figure 6 for very limited TA data! More information or explanation are required!
Author Response
1- The statement " This is due, at least in part, to the inaccuracies resulting from and even Se assimilation in the melt during inoculation in the hand ladle" was added in the text
2- We provided an answer in the reply to question number 1. We agrre with the reviewer that more information is required for a definitive conclusion as stated in the text in the sentence "To reach a definitive conclusion on this issue, more research data are required"
Reviewer 2 Report
The first version of the manuscript that I received in pdf format had many error messages (document attached) that made it difficult to understand the text. The current version is much clearer.

Author Response
Thank for your comments. We are glad that the issue was resolved